# First-Order Preconditioning via Hypergradient Descent

## Abstract

Standard gradient descent methods are susceptible to a range of issues that can impede training, such as high correlations and different scaling in parameter space. These difficulties can be addressed by second-order approaches that apply a preconditioning matrix to the gradient to improve convergence. Unfortunately, such algorithms typically struggle to scale to high-dimensional problems, in part because the calculation of specific preconditioners such as the inverse Hessian or Fisher information matrix is highly expensive. We introduce *first-order preconditioning* (FOP), a fast, scalable approach that generalizes previous work on hypergradient descent (Almeida et al., 1998; Maclaurin et al., 2015; Baydin et al., 2017) to *learn* a preconditioning matrix that only makes use of first-order information. Experiments show that FOP is able to improve the performance of standard deep learning optimizers on visual classification and reinforcement learning tasks with minimal computational overhead. We also investigate the properties of the learned preconditioning matrices and perform a preliminary theoretical analysis of the algorithm.

## 1 Introduction

High-dimensional nonlinear optimization problems often present a number of difficulties, such as strongly-correlated parameters and variable scaling along different directions in parameter space (Martens, 2016). Despite this, deep neural networks and other large-scale machine learning models applied to such problems typically rely on simple variations of gradient descent to train, which is known to be highly sensitive to these difficulties. While this approach often works well in practice, addressing the underlying issues directly could provide stronger theoretical guarantees, accelerate training, and improve generalization.

Adaptive learning rate methods such as Adam (Kingma and Ba, 2014), Adagrad (Duchi et al., 2011), and RMSProp (Tieleman and Hinton, 2012) provide some degree of higher-order approximation to re-scale updates based on per-parameter behavior. Newton-based methods make use of the curvature of the loss surface to both re-scale and rotate the gradient in order to improve convergence. Natural gradient methods (Amari, 1998) do the same in order to enforce smoothness in the evolution of the model's conditional distribution. In each of the latter two cases, the focus is on computing a specific linear transformation of the gradient that improves the conditioning of the problem. This transformation is typically known as a *preconditioning*, or *curvature*, matrix. In the case of quasi-Newton methods, the preconditioning matrix takes the form of the inverse Hessian, while for natural gradient methods it's the inverse Fisher information matrix. Computing these transformations is typically intractable for high-dimensional problems, and while a number of approximate methods exist for both (e.g., (Byrd et al., 1996; Martens and Grosse, 2015; Grosse and Martens, 2016)), they are often still too expensive for the performance gain they provide. These approaches also suffer from rigid inductive biases regarding the nature of the problems to which they are applied in that they seek to compute or approximate specific transformations. However, in large, non-convex problems, the optimal gradient transformation may be less obvious, or may even change over the course of training.

In this paper, we address these issues through a method we term *first-order preconditioning* (FOP). Unlike previous approaches, FOP doesn't attempt to compute a specific preconditioner, such as the inverse Hessian, but rather uses first-order hypergradient descent (Maclaurin et al., 2015) to *learn* an

---

**Algorithm 1** Learned First-Order Preconditioning (FOP)

---

1: **Require:** model parameters $\theta$, objective function $J$, FOP matrix $M$, learning rate $\epsilon$, hypergradient learning rate $\rho$
2: **for** t = 1,2,... **do**
3:      Draw data $x^{(t)}, y^{(t)} \sim \mathcal{D}$
4:      Perform forward pass: $\hat{y}^{(t)} = f_{\theta^{(t)}}(x^{(t)})$
5:      Compute loss $J(y^{(t)}, \hat{y}^{(t)})$
6:      Update inference parameters $\theta$: $\theta^{(t+1)} \leftarrow \theta^{(t)} - \epsilon M^{(t)} M^{(t)\top} \nabla_{\theta^{(t)}} J$
7:      Update preconditioning matrices:

$$M^{(t+1)} \leftarrow M^{(t)} + \rho\epsilon \left( \nabla_{\theta^{(t)}} J \left[ \nabla_{\theta^{(t-1)}} J \right]^\top + \nabla_{\theta^{(t-1)}} J \left[ \nabla_{\theta^{(t)}} J \right]^\top \right) M^{(t)}$$

8:      Cache $\nabla_{\theta^{(t)}} J$

---

adaptable transformation online directly from the task objective function. Our method adds minimal computational and memory cost to standard deep network training and results in improved convergence speed and generalization compared to standard approaches. FOP can be flexibly applied to any gradient-based optimization problem, and we show that when used in conjunction with standard optimizers, it improves their performance. To our knowledge, this is also the first successful application of any hypergradient method to the ImageNet dataset (Deng et al., 2009).

## 2 FIRST ORDER PRECONDITIONING

### 2.1 THE BASIC APPROACH

Consider a parameter vector $\theta$ and a loss function $J$. A traditional gradient update with a preconditioning matrix $P$ can be written as

$$\theta^{(t+1)} = \theta^{(t)} - \epsilon P^{(t)} \nabla_{\theta^{(t)}} J^{(t)}. \tag{1}$$

Our goal is to learn $P$. However, while we place no other constraints on our preconditioner, in order to ensure that it is positive semi-definite, and therefore does not reverse the direction of the gradient, we reparameterize $P$ as the product of a matrix $M$ with itself, changing Equation 1 to:

$$\theta^{(t+1)} = \theta^{(t)} - \epsilon M^{(t)} M^{(t)\top} \nabla_{\theta^{(t)}} J^{(t)}. \tag{2}$$

Under reasonable assumptions, gradient descent is guaranteed to converge with the use of even a random symmetric, positive-definite preconditioner, as we show in Appendix A.3. Because $\theta^{(t)}$ is a function of $M^{(t-1)}$, we can then backpropagate from the loss at iteration $t$ to the previous iteration's preconditioner via a simple application of the chain rule:

$$\frac{\partial J^{(t)}}{\partial M^{(t-1)}} = \frac{\partial J^{(t)}}{\partial \theta^{(t)}} \frac{\partial \theta^{(t)}}{\partial M^{(t-1)}}. \tag{3}$$

By applying the chain and product rules to Equation 2, the gradient with respect to the preconditioner is then simply

$$\nabla_{M^{(t-1)}} J^{(t)} = -\epsilon \left( \nabla_{\theta^{(t)}} J^{(t)} \left[ \nabla_{\theta^{(t-1)}} J^{(t-1)} \right]^\top + \nabla_{\theta^{(t-1)}} J^{(t-1)} \left[ \nabla_{\theta^{(t)}} J^{(t)} \right]^\top \right) M^{(t)}. \tag{4}$$

We provide a more detailed derivation in Appendix A.1. Note that ideally we would compute $\nabla_{M^{(t)}} J^{(t+1)}$ to update $M^{(t)}$, but as we don't have access to $J^{(t+1)}$ yet, we follow the example of Almeida et al. (1998) and assume that $J$ is smooth enough that this does not have an adverse effect on training. The basic approach is summarized in Algorithm 1. We use supervised learning as an example, but the same method applies to any gradient-based optimization. The preconditioned gradient can then be passed to any standard optimizer to produce an update for $M$. For example,

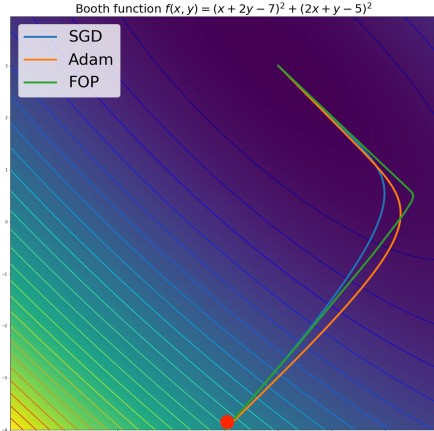 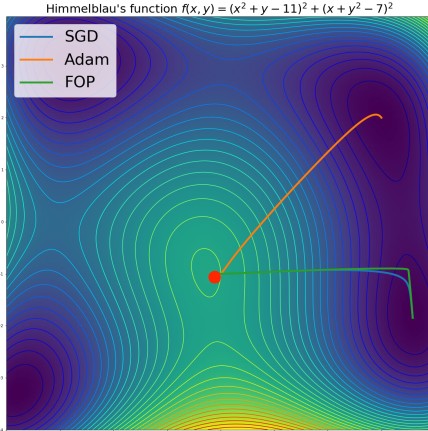

**Figure 1: A comparison of FOP to common optimizers on toy problems.** The red dot indicates the initial position on the loss surface. The purpose of these visualizations is not to establish the superiority of one optimizer over another, but rather to gain an intuition for their qualitative behavior. (Left) Gradient descent on the Booth function. FOP converges in 543 iterations, while SGD takes 832 steps and 6,221 for Adam. (Right) Gradient descent on Himmelbau's function. Adam, converging in 5398 iterations, finds a different global minimum from FOP and SGD, which converge in 289 and 386 steps, respectively. In both cases, we can see that FOP moves more aggressively across the objective function surface compared to the other methods. The poor performance of Adam is likely attributable to the non-stochastic nature of these toy settings.

we describe the procedure for using FOP with momentum (Polyak, 1964) in Section 2.3. For multi-layer networks, in order to make the simplest modification to normal backpropagation, we learn a separate $M$ for each layer, not a global curvature matrix.

To get an intuition for the behavior of FOP compared to standard algorithms, we observed its trajectories on a set of low-dimensional optimization problems (Figure 1). Interestingly, while FOP converged in fewer iterations than SGD and Adam (Kingma and Ba, 2014), it took more jagged paths along the objective function surface, suggesting that while it takes more aggressive steps, it is perhaps also able to change direction more rapidly.

## 2.2 SPATIAL PRECONDITIONING FOR CONVOLUTIONAL NETWORKS

In order to further reduce the computational cost of FOP in convolutional networks (CNNs), we implemented layer-wise *spatial* preconditioners, sharing the matrices across both input and output channels (results shown in Section 4). More concretely, if a convolutional layer has spatial kernels with shape $k \times k$, we can learn a preconditioner that is $k^2 \times k^2$. To implement this, when $\theta$ is a 4-tensor of kernels of shape $k \times k \times I \times O$, where $I$ and $O$ are the input and output channels, respectively, we can reshape it to a matrix of size $k^2 \times IO$, left-multiply it by the learned curvature matrix, and then reshape it back to its original dimensions. When $k$ is small, as is typically the case in deep CNNs, this preconditioner is small as well, resulting in both computational and memory efficiency. For large fully-connected networks, we adopt a low-rank preconditioner to save computation. In Appendix A.4, we offer further details and show that FOP is still effective even when the rank is very low.

## 2.3 FOP FOR MOMENTUM

FOP can be implemented alongside any standard optimizer, such as gradient descent with momentum (Polyak, 1964). Given a parameter vector $\theta$ and a loss function $J$, a basic momentum update with FOP matrix $M$ is typically expressed as two steps:

$$v^{(t+1)} = \alpha v^{(t)} + M^{(t)} M^{(t)\top} \nabla_{\theta^{(t)}} J \tag{5}$$

$$\theta^{(t+1)} = \theta^{(t)} - \epsilon v^{(t+1)}, \tag{6}$$

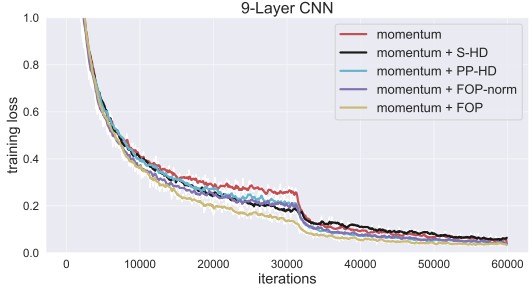

| Method | Test Accuracy | Adtl. Params | Adtl. Time (%) |
|---|---|---|---|
| momentum | $90.9 \pm 0.1$ | 0 | 0.0 |
| S-HD | $91.0 \pm 0.2$ | 9 | 0.2 |
| PP-HD | $91.3 \pm 0.3$ | $\approx 1.4M$ | 5.9 |
| FOP-norm | $91.3 \pm 0.1$ | 65 | 1.0 |
| FOP | $91.5 \pm 0.2$ | 65 | 0.8 |

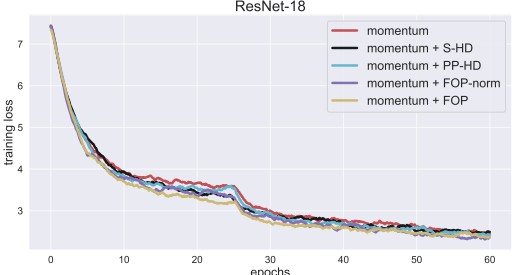

| Method | Test Accuracy | Adtl. Params | Adtl. Time (%) |
|---|---|---|---|
| momentum | $69.7 \pm 0.2$ | 0 | 0.0 |
| S-HD | $69.7 \pm 0.3$ | 22 | 0.3 |
| PP-HD | $69.8 \pm 0.2$ | $\approx 275M$ | 7.1 |
| FOP-norm | $69.9 \pm 0.3$ | 197 | 2.1 |
| FOP | $70.1 \pm 0.2$ | 197 | 1.7 |

**Figure 2: Results on CIFAR-10 (top) and ImageNet (bottom), averaged over 3 runs.** All models are trained with momentum as the base optimizer. We can see that FOP converges more quickly than standard and baseline methods, with slightly superior generalization performance. Learning a spatial curvature matrix adds negligible computational cost to the training process.

where $v$ is the velocity term and $\alpha$ is the momentum parameter. Combining Equations 5 and 6 allows us to write the full update as

$$\theta^{(t+1)} = \theta^{(t)} - \epsilon\alpha v^{(t)} - \epsilon M^{(t)} M^{(t)\top} \nabla_{\theta^{(t)}} J. \tag{7}$$

If, as in Maclaurin et al. (2015), we were meta-learning $M$ or only updating $M$ after a certain number of iterations, we would then have to backpropagate through $v$ to calculate the gradient for $M$. As we are updating $M$ online, however, we only need to calculate $\nabla_{M^{(t)}} J$. Therefore, the update is the same as for standard gradient descent. The experiments in Section 4 were performed using this modification of momentum.

## 3 RELATED WORK

Almeida et al. (1998) introduced the idea of using gradients from the objective function to learn optimization parameters such as the learning rate or a curvature matrix. However, their preconditioning matrix was strictly diagonal, amounting to an approximate Newton algorithm (Martens, 2016), and they only tested their framework on simple optimization problems with either gradient descent or SGD. This is related to common deep learning optimizers such as Adagrad (Duchi et al., 2011) and Adam (Kingma and Ba, 2014), which also learn adaptive per-parameter learning rates, but unlike FOP, do not induce a rotation on the gradient. As we note in Section 2.3, an advantage of FOP is that it may also be used in conjunction with any such optimizer. More recently, Maclaurin et al. (2015) applied the backpropagation-across-iterations approach in a neural network context, terming the process *hypergradient descent*. Their method backpropagates through multiple iterations of the training process of a relatively shallow network to meta-learn a learning rate. However, this method can incur significant memory and computational cost for large models and long training times. Baydin et al. (2017) instead proposed an online framework directly inherited from Almeida et al. (1998) that used hypergradient-based optimization in which the learning rate is updated after each iteration. Our method extends this idea to not only learn existing optimizer parameters (e.g., learning rate, momentum), but to introduce novel ones in the form of a non-diagonal, layer-specific preconditioning matrix for the gradient. It's also important to discuss the relationship of FOP to other, non-hypergradient preconditioning methods for deep networks. These mostly can be sorted into one

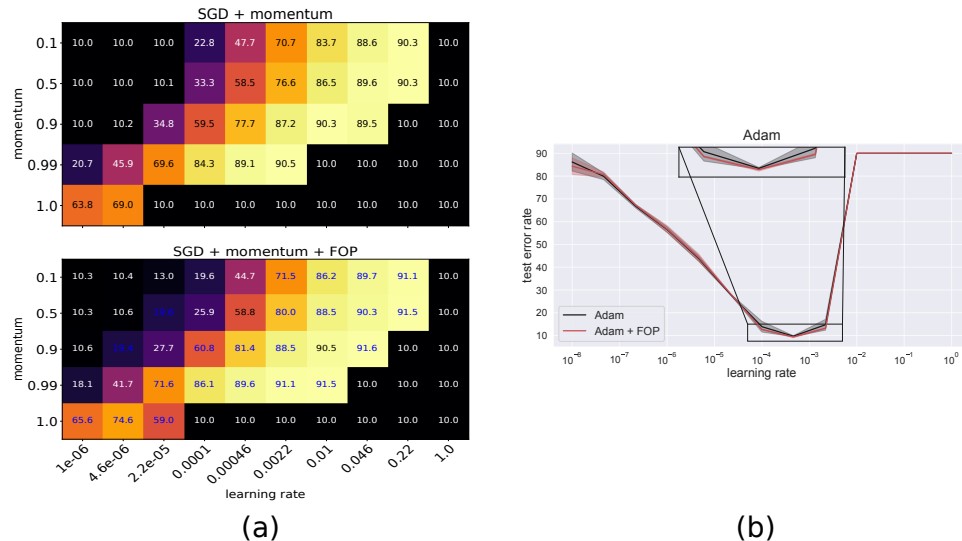

(a)                                                                 (b)

**Figure 3: Adding FOP improves the hyperparameter robustness of standard optimizers.** (a) The final test accuracy of a 9-layer CNN trained on CIFAR-10 for different settings of SGD with momentum (top) and SGD with momentum and FOP (bottom), averaged over three runs. Settings in which adding FOP improves performance by at least one standard deviation are highlighted in blue. FOP appears to be most useful for higher values of the learning rate and momentum parameters. The FOP matrices were trained with Adam with a learning rate of $5 \times 10^{-4}$. (b) The performance of Adam for a range of learning rates both with and without FOP, averaged over three runs. While performance is similar, the top performing models are improved by the addition of FOP. The FOP matrices were trained using the same settings as the models in (a).

of two categories, quasi-Newton algorithms and natural gradient approaches. Quasi-Newton methods seek to learn an approximation of the inverse Hessian. L-BFGS, for example, does this through tracking the differences between gradients across iterations (Byrd et al., 1996). This is significantly different from FOP, although the outer product of gradients used in the update for FOP can also be an approximation of the Hessian or Gauss-Newton matrices (Bottou et al., 2016). Natural gradient methods, such as K-FAC (Martens and Grosse, 2015) and KFC (Grosse and Martens, 2016), which approximate the inverse Fisher information matrix, bear a much stronger resemblance to FOP. However, there are notable differences. First, unlike FOP, these methods perform extra computation to ensure the invertibility of their curvature matrices. Second, the learning process for the preconditioner in these methods is completely different, as they do not backpropagate across iterations.

# 4    EXPERIMENTS

We measured the performance of FOP on visual classification and reinforcement learning tasks. In order to measure the importance of the rotation induced by the preconditioning matrices in addition to the scaling, we also implemented hypergradient descent (HD) methods to learn a scalar layer-wise learning rate (S-HD) and a per-parameter (PP-HD) learning rate. The former is the same method implemented by Baydin et al. (2017), and the latter is equivalent to a strictly diagonal curvature matrix. We also implement a method we call normalized FOP (FOP-norm), in which we re-scale the preconditioning matrix to avoid any effect on the learning rate and rely solely on standard learning rate settings. Further details on this method can be found in Appendix A.2. All experiments were implemented using the TensorFlow library (Abadi et al., 2015)[1].

---

[1]Code currently available at this link:
https://drive.google.com/file/d/1vhB4fxDuxaYJcNP6ioEJQf4CLHxhy1ka/view?usp=sharing.

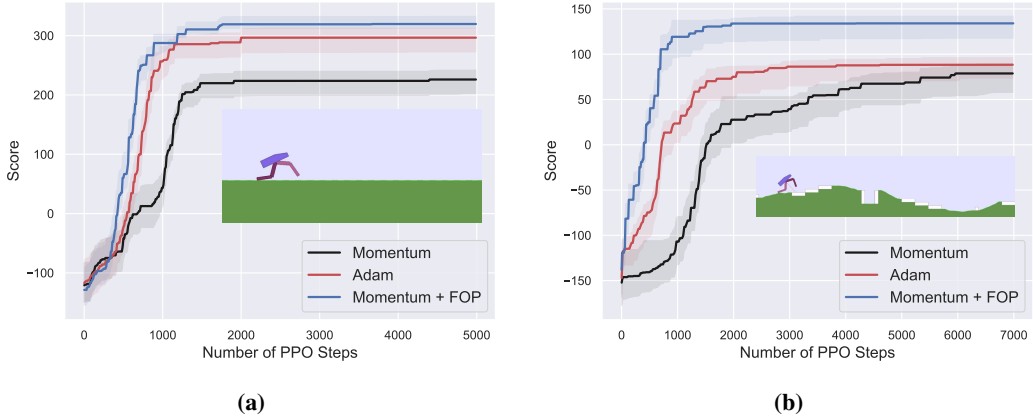

**Figure 4: Bipedal Walker on varying terrains.** We plot the the median reward (of the best seen policy so far) per PPO step across 10 runs for (a) and 5 runs for (b), respectively. The shading denotes the 95% bootstrapped confidence intervals. Momentum with FOP outperforms momentum alone significantly, both in speed of learning and final score, in addition to beating Adam.

### 4.1 CIFAR-10

For CIFAR-10 (Krizhevsky, 2009), an image dataset consisting of 50,000 training and 10,000 test $32 \times 32$ RGB images divided into 10 object classes, we implemented a 9-layer convolutional architecture inspired by Springenberg et al. (2014). We trained each model for 150 epochs with a batch size of 128 and initial learning rate 0.05, decaying the learning rate by a factor of 10 after 80 epochs. For S-HD, PP-HD, and FOP, we use Adam as the hypergradient optimizer with a learning rate of 1e-4. The results are plotted in Figure 2. FOP produces a significant speed-up in training and improves final test accuracy compared to baseline methods, including FOP-norm, indicating that both the rotation and the scaling learned by FOP is useful for learning.

### 4.2 IMAGENET

The ImageNet dataset consists of 1,281,167 training and 50,000 validation $299 \times 299$ RGB images divided into 1,000 categories (Deng et al., 2009). Here, we trained a ResNet-18 (He et al., 2015) model for 60 epochs with a batch size of 256 and an initial learning rate of 0.1, decaying by a factor of 10 at the 25th and 50th epochs. A summary of our results is displayed in Figure 2. We can see that the improved convergence speed and test performance observed on CIFAR-10 is maintained on this deeper model and more difficult dataset.

### 4.3 HYPERPARAMETER ROBUSTNESS

In addition to measuring peak performance, we also tested the effect of FOP on the robustness of standard optimizers to hyperparameter selection. Baydin et al. (2017) demonstrated the ability of scalar hypergradients to mitigate the effect of the initial learning rate on performance. We therefore tested whether this benefit was preserved by FOP, as well as whether it extended to other hyperparameter choices, such as the momentum coefficient. Our results, summarized in Figure 3, support this idea, showing that FOP can improve performance on a wide array of optimizer settings. For momentum (Fig. 3a), we see that the performance gap is greater for higher learning rates and momentum values, and in several instances FOP is able to train successfully where pure SGD with momentum fails. For Adam, the difference is smaller, although in the highest-performance learning rate region adding FOP to Adam outperforms the standard method. We hypothesize that this smaller difference is in large part due to unanticipated effects that preconditioning the gradient has on the adaptive moment estimation performed by Adam. We leave further investigation into this interaction for future work.

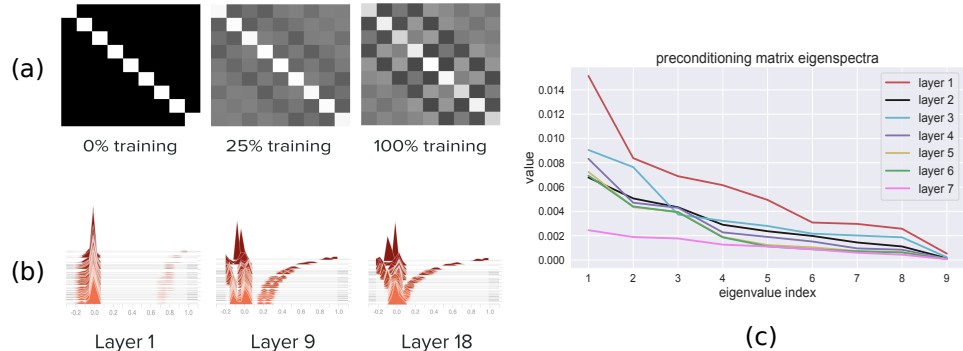

**Figure 5: Understanding the learned preconditioning matrices ($MM^\top$) for CIFAR-10.** (a) The evolution of an example preconditioning matrix throughout the training process from the ninth layer in a ResNet-18 model trained on ImageNet. Each layer learned a similar whitening structure. (b) The histograms of matrix values across layers during training. The training process is traced by going from back to front in the plots. We can see that the convergence of the values of the matrix, corresponding to a stronger decorrelation structure and a reduced $L_2$ norm, is stronger in the higher layers of a network. (c) The sorted eigenvalues of the final learned preconditioning matrices for the first seven layers of a 9-layer network trained on CIFAR-10 (the top two layers were $1 \times 1$ kernels). The distribution shifts downward and becomes more uniform in higher layers. This is interesting, as while a uniform distribution of eigenvalues is considered helpful in aiding convergence, the downward shift in values makes the matrix less invertible.

### 4.4 REINFORCEMENT LEARNING

Reinforcement learning (RL) tasks are often more challenging than supervised learning, since the rewards in RL are sparse and the loss-landscape is frequently deceptive (Sutton and Barto, 2018). We evaluated the performance of FOP on one such RL task - the training of a 2D bipedal robot walker (Brockman et al., 2016; Klimov, 2016) in two different terrains (see the inserts in Figures 4 (a) and (b) for illustration). This 2D robotic domain has been used in recent literature as a benchmark and for validation of new RL algorithms (Ha, 2017; 2018; Song et al., 2018; Wang et al., 2019). The agent is trained to move forward (from left to right) in a given terrain, within a time limit of maximum 2000 steps and without falling over. The controller for the agent consists of simple feedforward policy and value networks. Both the networks are trained using the Proximal Policy Optimization (PPO) algorithm (Schulman et al., 2017) (see Appendix A.5 for details).

Figures 4 (a) and (b) plot the bipedal walker scores obtained from training the controller in two different terrains. For both terrains, FOP with momentum trains faster and achieves better asymptotic performance than baseline methods (SGD with momentum and Adam), thus demonstrating the effectiveness of FOP in RL tasks. We hypothesize that FOP's advantage results in part from deceptive loss surfaces, since FOP's ability to quickly change direction (Figure 1) may help it avoid getting stuck in local optima (see Appendix A.5 for a discussion of deceptiveness in this task).

## 5 WHAT IS FOP LEARNING?

By studying the learned preconditioning matrices, it's possible to gain an intuition for the effect FOP has on the training process. Interestingly, we found visual tasks induced similar structures in the preconditioning matrices across initializations and across layers. Visualizing the matrices (Figure 5a) shows that they develop a decorrelating, or *whitening*, structure: each of the 9 positions in the 3x3 convolutional filter sends a strong positive weight to itself, and negative weights to its immediate neighbors, without wrapping over the corners of the filter. This is interesting, as images are known to have a high degree of spatial autocorrelation (Barlow, 1961; 1989). As a mechanism for reducing redundant computation, whitening visual inputs is known to be beneficial for both retinal processing in the brain (Atick and Redlich, 1992; Van Hateren, 1992) and in artificial networks (Desjardins et al., 2015; Huang et al., 2018). However, it is more unusual to consider whitening of the learning signal, as observed in FOP, rather than the forward activities.

This learned pattern is accompanied by a shift in the norm of the curvature matrix, as the diagonal elements, initialized to one, shrink in value, and the off-diagonal elements grow. This shift in distribution is visualized in Figure 5b, and grows stronger deeper in the network. It is possible that a greater degree of decorrelation is helpful for kernels that must disentangle high-level representations.

We can also examine the eigenvalue spectra for the learned matrices across layers (Figure 5c). We can see that the basic requirement that in general a preconditioning matrix must be positive semi-definite, so as not to reverse the direction of the gradient, is met. However, the eigenvalues are very small in magnitude, indicating a near-zero determinant. This results in a matrix that is essentially non-invertible, an interesting property, as quasi-Newton and natural gradient methods seek to compute or approximate the inverse of either the Hessian or Fisher information matrix. The implications of this non-invertibility are avenues for future study. Furthermore, the eigenvalues grow smaller, and their distribution more uniform, higher in the network, in accordance with the pattern observed in Figure 5b. A uniform eigenspectrum is seen as an attribute of an effective preconditioning matrix (Li, 2015), as it indicates an even convergence rate in parameter space.

## 6 CONVERGENCE

Our experiments indicate that the curvature matrices learned by FOP converge to a relatively fixed norm roughly two-thirds of the way through training (Figure 5b). This is important, as it indicates that the effective learning rate induced by the preconditioners stabilizes. This allows us to perform a preliminary convergence analysis of the algorithm in a manner analogous to Baydin et al. (2017).

Consider a modification of FOP in which the symmetric, positive semi-definite preconditioning matrix $P^{(t)} = M^{(t)}M^{(t)\top}$ is rescaled at each iteration to have a certain norm $\gamma^{(t)}$, such that $\gamma^{(t)} \approx ||P^{(t)}||_2$ when $t$ is small and $\gamma^{(t)} \approx p^{(\infty)}$ when $t$ is large, where $p^{(\infty)}$ is some chosen constant. Specifically, as in Baydin et al. (2017), we set $\gamma^{(t)} = \delta(t)||P^{(t)}||_2 + (1 - \delta(t))p^{(\infty)}$, where $\delta(t)$ is some function that decays over time and starts training at 1 (e.g., $1/t^2$).

This formulation allows us to extend the convergence proof of Baydin et al. (2017) to FOP, under the same assumptions about the objective function $J$:

**Theorem 1** *Suppose that $J$ is convex and $L$-Lipschitz smooth with $\|\nabla_\theta J\| < K$ for some fixed $K$ and all model parameters $\theta$. Then $\theta_t \to \theta^*$ if $p^{(\infty)} < 1/L$ and $t\delta(t) \to 0$ as $t \to \infty$, where the $\theta_t$ are generated by (non-stochastic) gradient descent.*

*Proof.* We can write

$$\|P^{(t)}\| \leq \|P^{(0)}\| + \rho \sum_{i=0}^{t-1} \left|[\nabla_{\theta_{i+1}} J]^\top \nabla_{\theta_i} J\right| \leq \|P^{(0)}\| + \rho \sum_{i=0}^{t-1} \|\nabla_{\theta_{i+1}} J\| \|\nabla_{\theta_i} J\| \leq \|P^{(0)}\| + t\rho K^2,$$

where $\rho$ is the hypergradient learning rate. Our assumptions about the limiting behavior of $t\delta(t)$ then imply that $\delta(t)\|P^{(t)}\| \to 0$ and so $\gamma^{(t)} \to p^{(\infty)}$ as $t \to \infty$. For sufficiently large $t$, we therefore have $\gamma^{(t)} \in (\frac{1}{L+1}, \frac{1}{L})$. Note also that as $P$ is symmetric and positive semi-definite, it will not prevent the convergence of gradient descent (Appendix A.3). Moreover, preliminary investigation shows that the angle of rotation induced by the FOP matrices is significantly below $90°$ (Appendix Figure A.1). Lillicrap et al. (2016) showed that such a rotation does not impede the convergence of gradient descent. Because standard SGD converges under these conditions (Karimi et al., 2016), FOP must as well.

## 7 CONCLUSION

In this paper, we introduced a novel optimization technique, FOP, that learns a preconditioning matrix online to improve convergence and generalization in large-scale machine learning models. We tested FOP on several problems and architectures, examined the nature of the learned transformations, and provided a preliminary analysis of FOP's convergence properties (Appendix 6). There are a number of opportunities for future work, including learning transformations for other optimization parameters (e.g., the momentum parameter in case of the the momentum optimizer), expanding and generalizing our theoretical analysis, and testing FOP on a wider variety of models and data sets.

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

## A  SUPPLEMENTAL INFORMATION

### A.1  DERIVATION OF UPDATE FOR $M^{(t)}$

Our update rule for the model parameter $\theta$, given by Equation 2, is

$$\theta^{(t)} = \theta^{(t-1)} - \epsilon M^{(t-1)} M^{(t-1)\top} \nabla_{\theta^{(t-1)}} J^{(t-1)}. \tag{8}$$

Then by applying the the chain and product rules the update is given by

$$\nabla_{M^{(t-1)}} J^{(t)} = \nabla_{\theta^{(t)}} J^{(t)\top} \left( -\epsilon M^{(t-1)} \nabla_{\theta^{(t-1)}} J^{(t-1)} \right)$$

$$+ \nabla_{\theta^{(t)}} J^{(t)} \left( -\epsilon \nabla_{\theta^{(t-1)}} J^{(t-1)\top} M^{(t-1)\top} \right) \tag{9}$$

$$= -\epsilon \left( \nabla_{\theta^{(t)}} J^{(t)} \left[ \nabla_{\theta^{(t-1)}} J^{(t-1)} \right]^{\top} + \nabla_{\theta^{(t-1)}} J^{(t-1)} \left[ \nabla_{\theta^{(t)}} J^{(t)} \right]^{\top} \right) M^{(t)}. \tag{10}$$

### A.2  NORMALIZED FOP

In order to control for the scaling induced by FOP and measure the effect of its rotation only, we introduce *normalized* FOP, which performs the following parameter update:

$$\theta^{(t+1)} = \theta^{(t)} - \epsilon \sqrt{n} \frac{M^{(t)} M^{(t)\top}}{\|M^{(t)} M^{(t)\top}\|} \nabla_{\theta^{(t)}} J, \tag{11}$$

where $n$ is the first dimension of $M$. This update has the effect of normalizing the preconditioner $M^{(t)} M^{(t)\top}$, then re-scaling the update by $\sqrt{n}$ to match the norm of gradient descent, as $\|I_n\| = \sqrt{n}$, where $I_n$ is the identity matrix of size $n$.

### A.3  CONVERGENCE OF PRECONDITIONED GRADIENT DESCENT

We demonstrate that given a random, symmetric positive semi-definite preconditioning matrix $P$, gradient descent will still converge at a linear rate, modeling our proof after that of Karimi et al. (2016).

**Theorem 2** *Consider a convex, L-Lipschitz objective function $f(\theta)$ with global minimum $f^*$ which obeys the Polyak-Łojasiewicz (PL) Inequality (Polyak, 1963),*

$$\frac{1}{2}\|\nabla_\theta f\|^2 \geq \mu(f(\theta) - f^*), \quad \forall \theta. \tag{12}$$

*Then applying the gradient update method given by*

$$\theta_{k+1} = \theta_k - \rho P \nabla_{\theta_k} f, \tag{13}$$

*where $P$ is a real, symmetric positive semi-definite matrix and $\rho = \frac{2\lambda_{\min} - \lambda_{\max}^2}{L}$ is the step size, where $\lambda_{\min}$ and $\lambda_{\max}$ are the minimum and maximum eigenvalues of $P$, respectively, results in a global linear convergence rate given by*

$$f(\theta_k) - f^* \leq (1 - \mu\rho)^k (f(\theta_0) - f^*). \tag{14}$$

*Proof.* First, assume a step-size of $1/L$. Given that $f$ is $L$-Lipschitz continuous, we can write

$$f(\theta_{k+1}) - f(\theta_k) \leq \langle \nabla f, \theta_{k+1} - \theta_k \rangle + \frac{L}{2}\|\theta_{k+1} - \theta_k\|^2.$$

where $\nabla f$ denotes $\nabla_{\theta_k} f$. Plugging in the gradient update equation gives

$$f(\theta_{k+1}) - f(\theta_k) \leq \langle \nabla f, -\frac{1}{L} P \nabla f \rangle + \frac{L}{2}\| -\frac{1}{L} P \nabla f \|^2$$

$$= -\frac{1}{L}(\nabla f)^\top P \nabla f + \frac{1}{2L}\|P \nabla f\|^2. \tag{15}$$

Let $Q^\top \Lambda Q = P$ be the eigendecomposition of $P$, such that the columns of $Q$ are the orthonormal eigenvectors and $\Lambda$ is a diagonal matrix whose entries are the eigenvalues of $P$, which are all non-negative due to symmetric positive semi-definiteness of $P$. We can then rewrite the first term of Equation 13 as

$$-\frac{1}{L}(\nabla f)^\top Q^\top \Lambda Q \nabla f.$$

We now change our basis, letting $G = Q\nabla f$ and define $\lambda_{\min} = \min_i\{\lambda_i\}$ and $\lambda_{\max} = \max_i\{\lambda_i\}$, where $\lambda_i$ are the eigenvalues. Then we have

$$-\frac{1}{L}(\nabla f)^\top Q^\top \Lambda Q \nabla f = -\frac{1}{L}G^\top \Lambda G \le -\frac{1}{L}\lambda_{\min} G^\top G = -\frac{1}{L}\lambda_{\min}\|G\|^2. \tag{16}$$

Examining the second term of Equation 13, we have

$$\frac{1}{2L}\|P\nabla f\|^2 = \frac{1}{2L}(P\nabla f)^\top P\nabla f$$

$$= \frac{1}{2L}(\nabla f)^\top Q^\top \Lambda Q Q^\top \Lambda Q \nabla f$$

$$= \frac{1}{2L}G^\top \Lambda^2 G$$

$$\le \frac{1}{2L}\lambda_{\max}^2 G^\top G$$

$$= \frac{1}{2L}\lambda_{\max}^2\|G\|^2. \tag{17}$$

Combining the results of Equations 14 and 15 gives

$$f(\theta_{k+1}) - f(\theta_k) \le -\frac{1}{L}\lambda_{\min}\|G\|^2 + \frac{1}{2L}\lambda_{\max}^2\|G\|^2$$

$$= -\frac{1}{L}\left(\lambda_{\min} - \frac{\lambda_{\max}^2}{2}\right)\|G\|^2. \tag{18}$$

We can then revert to the original basis:

$$\|G\|^2 = (Q\nabla f)^\top Q\nabla f = (\nabla f)^\top Q^\top Q \nabla f = (\nabla f)^\top \nabla f = \|\nabla f\|^2,$$

giving us

$$f(\theta_{k+1}) - f(\theta_k) \le -\frac{1}{L}\left(\lambda_{\min} - \frac{\lambda_{\max}^2}{2}\right)\|\nabla f\|^2. \tag{19}$$

By the PL inequality we have that $\|\nabla f\|^2 \ge 2\mu(f(\theta_k) - f^*)$ for some $\mu > 0$. Plugging this in gives

$$f(\theta_{k+1}) - f(\theta_k) \le -\frac{\mu}{L}\left(2\lambda_{\min} - \lambda_{\max}^2\right)(f(\theta_k) - f^*) \tag{20}$$

Then let $\rho = \frac{1}{L}\left(2\lambda_{\min} - \lambda_{\max}^2\right)$. We have

$$f(\theta_{k+1}) - f(\theta_k) \le -\mu\rho f(\theta_k) + \mu\rho f^*.$$

Rearranging and subtracting $f^*$ from both sides gives

$$f(\theta_{k+1}) - f^* \le (1 - \mu\rho)(f(\theta_k) - f^*). \tag{21}$$

Applying Equation 21 recursively gives the desired convergence:

$$f(\theta_{k+1}) - f^* \le (1 - \mu\rho)^{k+1}(f(\theta_0) - f^*). \tag{22}$$

Thus this preconditioned gradient descent converges with a step size $\frac{2\lambda_{\min} - \lambda_{\max}^2}{L}$. Standard gradient descent converges with a step size $1/L$ under these assumptions. We also note that even if $P$ changes over the course of training, the required step-size will vary, but convergence will still occur.

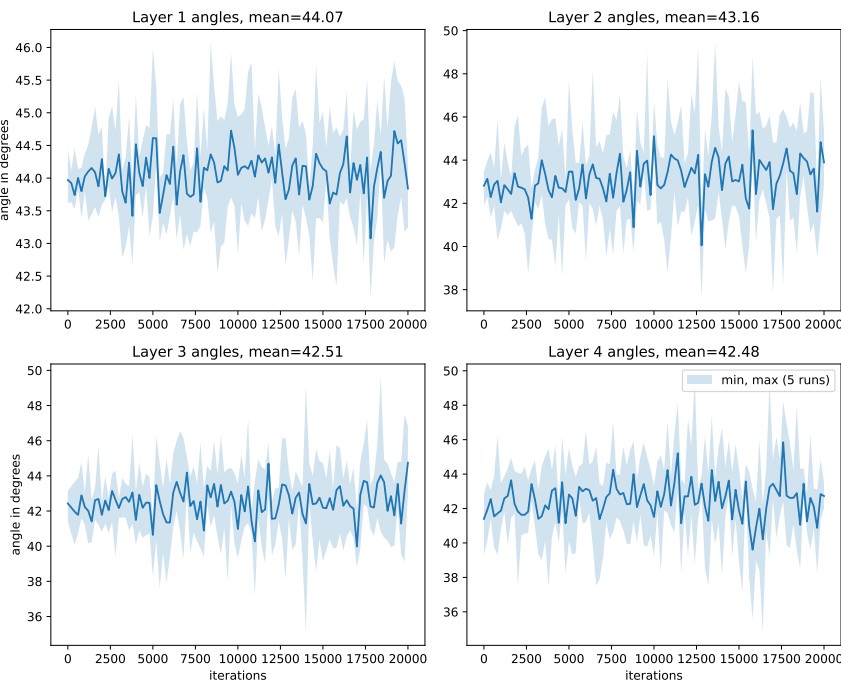

**Figure A.1: Angles in degrees between $\nabla J^{(t)}$ and $(I + M^{(t)}M^{(t)\top})\nabla J^{(t)}$ for each layer of a 4 layer fully connected network on MNIST.** Values are averaged over 5 runs and displayed every 200 iterations. Shaded areas represent the min and max values over all the runs, showing that the angle only varies slightly. Notice that the angle is usually slightly below $45°$, indicating that the while FOP does induce a rotation, it is far from orthogonal to the vanilla learning signal.

### A.4 Low-Rank FOP

If $\theta$ is an $m \times n$ matrix, and we only apply the preconditioning matrix over input dimensions (and share it across output dimensions), then a full-rank $M$ would necessarily be $m \times m$. When $m$ is large, preconditioning the gradient becomes expensive. Instead, we can apply a rank-$k$ $M$, with $M \in \mathbb{R}^{m \times k}$ and $k < m$. To ensure stable performance at the beginning of training, we initialize the preconditioning matrix as close as possible to the identity matrix, even for a low rank $M$, so that the algorithm begins as straightforward gradient descent and learns to depart from vanilla SGD over time. Thus, we set the preconditioner $P$ to be

$$P = I_m + MM^\top,\tag{23}$$

where $I_m$ is the $m \times m$ identity matrix and $M_{ij} \sim \mathcal{N}(0, \sigma^2)$, where $\sigma^2$ is small so that $P$ starts out close to the identity matrix. This effectively decomposes the update into an isotropic scaling of the gradient (multiplication by $I_m$ scaled by the learning rate) and a rotation (multiplication by $MM^\top$). We tested the effect of rank on a simple fully-connected network trained on the MNIST dataset (LeCun and Cortes, 2010). The results, shown in Figure A.2, indicate that FOP is able to accelerate training compared to standard SGD (with momentum 0.9) with all values of $k$ and improve final test accuracy even with fairly low values of $k$.

### A.5 Experimental Setup for Reinforcement Learning

Illustrated in the inserts of Figure 4 (a) and (b), the bipedal robot is made of a hull and two legs, and each leg consists of a hip and a knee controlled by motor joints, respectively. The action space therefore has 4 dimensions, each corresponding to the torque applied at each joint, respectively. The state space has 24 state variables, which includes readings from 10 LIDAR rangefinders and 14 other state variables about the robot itself, such as the angle and angular velocity of the hull, horizontal and vertical speed of the body, position and angular speed of joints, legs contact with ground, etc. The agent is trained to move forward (from left to right) in a given terrain, within a

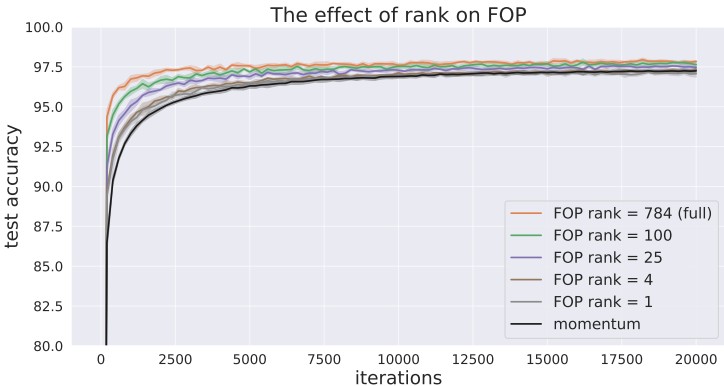

**Figure A.2: FOP is able to improve training even with very low ranks.** We plot the test accuracy over the course of training for a 4-layer fully-connected network with 100 units per layer trained on MNIST for different FOP matrix ranks, averaged over 5 runs each. The gradient was preconditioned by $I_m + MM^\top$, where $M$ was rank $k$. 784 is full rank for the first layer, and 100 is full rank for all layers except for the first. The larger $k$ is, the better the performance, both in terms of speed and final accuracy, compared to vanilla SGD with momentum 0.9. For $k < 4$, final accuracy is no longer better, but the initial training remains slightly faster. Final test accuracy of the full rank matrix is better than baseline test accuracy by 0.6% with $p$-value $< $ 1e-4.

time limit of maximum 2000 steps and without falling over. The reward at each time step is the distance traveled forward in that step, with penalty terms that consist of applied torques and hull angle from horizontal axis. If the agent falls over, the reward for that step will be -100, and the episode terminates immediately. This asymmetry in positive and negative rewards makes the task deceptive, since the agents could learn to move slowly or even move backward or stand still to avoid a large penalty from falling down. For each experiment in a given terrain, while the landscape remains fixed, both the initial position of the agent and the initial force applied to its body are randomly initialized.

We adopt the Proximal Policy Optimization (PPO) algorithm (Schulman et al., 2017), more specifically, PPO-Clip, a popular and competitive RL algorithm based on the implementation available in OpenAI Baselines (Dhariwal et al., 2017). The controller consists of a policy network and a value network. The policy network is a 3-layer feedforward neural network with input size of 24, hidden size of 40 ($tanh$ activation), and output size of 4 (bounded between -1 and 1). The value network shares the input and the hidden layers with the policy network and it has a separate fully-connected layer that connects to the value output. Note that we apply FOP with full rank to all the layers including both policy and value layers. We did hyperparamter search for the learning rate in the range of $[1e - 4, 1e - 3]$ and minibatch size in the range of $[8, 64]$ for each of FOP with momentum and the two baseline methods (SGD with momentum and Adam). The best hyperparameter setting for each method was adopted to run experiments and plot the results in Figure 4. Specifically, the best learning rate setting for FOP with momentum, Adam, and SGD with momentum were $3e - 4$, $3e - 4$ and $3.5e - 4$, respectively, and the best minibatch size setting was 32 for all three methods.

