# OpenReview forum: "First-Order Preconditioning via Hypergradient Descent"
_ICLR.cc/2020/Conference — Reject_

### Official Review · AnonReviewer2 · 2019-10-21
**Official Blind Review #2**

**Rating:** 3

**Review:**

This paper proposes an interesting optimization algorithm called first-order preconditioning (FOP).
The basic idea of FOP is updating the preconditioned matrix by its gradient, which avoid calculating or approximating the Hessian directly. To make the algorithms more practical, the authors also conduct the low-rank FOP and the momentum-type version. The empirical studies on CIFAR-10 and ImageNet validate the effectives of the proposed algorithms.

Major comments:

1. Section 2.1 says “we follow the example of Almeida et al. (1998) and assume that J does not dramatically”. However, the goal of FOP is to encourage J reduce faster. Is there any conflict?

2. In low-rank FOP, the initial preconditioner P contains the term I_m which does not exist in standard FOP (section 2.1). How does this term affect the update procedure? Can you provide some details?

3. Theorem 2 provide a linear convergence of FOP under convex, Lipschitz and PL condition. The proof relaxes the preconditioner P into its minimum and maximum eigenvalues. Since P changes over the course of training, it is difficult to check weather the result of Theorem 2 is stronger than gradient descent method.

4. Why the experimental results not include the other second order optimization algorithms such as K-FAC and KFC?

Minor comment:

The notations M in (1) (2) and (5) are ambiguous. It is prefer to use another letter to present the preconditioner in (1).


**Experience Assessment:**

I have published one or two papers in this area.

**Review Assessment: Checking Correctness Of Derivations And Theory:**

I carefully checked the derivations and theory.

**Review Assessment: Checking Correctness Of Experiments:**

I assessed the sensibility of the experiments.

**Review Assessment: Thoroughness In Paper Reading:**

I read the paper thoroughly.

---

> ### Author Response · Authors · 2019-11-08
> **Response to Reviewer 2**
>
> Thank you very much for your comments, as well as for noting the novelty of FOP and its strong empirical performance! We’d also like to clarify that part of FOP’s novelty is that it doesn’t in any way attempt to approximate the Hessian--it learns an arbitrary linear transformation of the gradient directly from the task loss. We believe we can address your concerns:
>
> 1. “Section 2.1 says “we follow the example of Almeida et al. (1998) and assume that J does not dramatically”. However, the goal of FOP is to encourage J reduce faster. Is there any conflict?”
>       - This is a good question! Almeida et al. are simply assuming that the objective function is relatively smooth. We make the same assumption, with the simple goal of moving along this smooth surface in fewer steps compared to other algorithms--these features are complementary to one another. We will certainly clarify our language on this point!
>
> 2. “In low-rank FOP, the initial preconditioner P contains the term I_m which does not exist in standard FOP (section 2.1). How does this term affect the update procedure? Can you provide some details?”
>       - Good question! I_m is introduced to encourage a more diagonal preconditioner in early training, so that FOP approximates SGD while the preconditioner is first adapting from its initialization. It effectively decomposes the update into an isotropic scaling of the gradient (multiplication by I_m x the learning rate) and a rotation (multiplication by MM^T). We will add these details to the paper!
>
> 3. “Theorem 2 provide a linear convergence of FOP under convex, Lipschitz and PL condition. The proof relaxes the preconditioner P into its minimum and maximum eigenvalues. Since P changes over the course of training, it is difficult to check weather the result of Theorem 2 is stronger than gradient descent method.”
>       - Thank you for noting this! The goal of Theorem 2 isn’t to prove that FOP converges more quickly than standard gradient descent (that we leave to future work), but rather simply that the algorithm is still guaranteed to converge even with an arbitrary preconditioner. We also note at the end of the proof that P changing does not affect the proof itself, it simply implies that the step-size must be adaptive to maintain the guarantee of convergence. We refer to Karimi et al. (2016) [1] for further details.
>
> 4. “Why the experimental results not include the other second order optimization algorithms such as K-FAC and KFC?”
>       - Good question! We’d like to note that FOP is not a second-order method--it utilizes only first-order information. This is one of the primary distinctions between FOP and previous preconditioning methods like K-FAC and KFC, in addition to learning an arbitrary preconditioner directly from the loss. Therefore, we found it more appropriate to compare FOP to other adaptive first-order hypergradient methods. To get a sense of the comparison to K-FAC and KFC on CIFAR-10, for example, we recommend viewing Figure 3 of the KFC paper. Importantly, FOP maintains its performance advantage over SGD at convergence, while SGD eventually overtakes KFC. FOP’s efficiency also enables to scale more easily to larger problems like ImageNet.
>
> 5. “The notations M in (1) (2) and (5) are ambiguous. It is prefer to use another letter to present the preconditioner in (1).”
>       - This is a very good point! We agree, and will change M to P in Eq. 1, as well as note that we set P = MM^T for Eq. 2.
>
> Thank you very much once again for helping us to improve our paper, and let us know if there’s anything else we can do to address your concerns! If you have any follow-up thoughts or questions, we'll be happy to reply again.
>
> [1] https://arxiv.org/abs/1608.04636

---

> > ### Comment · AnonReviewer2 · 2019-11-12
> > **Thanks for your feedback**
> >
> > The points 1, 2 and 5 in the response are OK. I’m a little disappointed that Theorem 2 can not show the algorithm is faster than gradient descent. I agree SGD eventually overtakes KFC, however, the approximation of fisher matrix in KFC is only dependent on first-order information, which is similar to FOP. In summary, I decide to keep my rating.

---

> > > ### Author Response · Authors · 2019-11-15
> > > **Thank you!**
> > >
> > > Thank you very much once again for the feedback!
> > >
> > > 1) It’s true that we do not provide any proofs showing faster convergence—that’s certainly something we’d like to do for future work. However, we do empirically demonstrate faster convergence. We believe our new results on reinforcement learning further underscore the advantages of FOP.
> > >
> > > 2) We do not experimentally compare FOP with KFC, though we agree with the reviewer’s point that it is likely similar enough to warrant a direct comparison of performance.  However, we’d like to point out, as the reviewer notes, that KFC is overtaken by SGD at convergence, an issue from which FOP does not suffer in our experiments. We’d also like to highlight a few of the broader contributions and novelty of the paper, specifically the introduction of a non-invertible preconditioning matrix learned via hypergradients and the first application of online hypergradient optimization to ImageNet and reinforcement learning, both with strong performance. We believe these contributions may be of interest to the community at large.

---

### Official Review · AnonReviewer3 · 2019-10-24
**Official Blind Review #3**

**Rating:** 3

**Review:**

This paper studies hypergradient descent for precondition matrices. The goal is to learn an adaptable preconditioning for the task while training. Specifically, they take the gradient of the loss wrt the precondition matrix and update the precondition matrix to decrease the loss. They reparametrize the precondition matrix to ensure it is positive-definite and provide low-rank approximations and they provide cheap approximations for CNNs.

Pros:
- Figure 3 and 4 show promising results on cifar10 with a 9-layer cnn.
- Figure 4 shows FOP can improve the accuracy for particular hyper-parameters. In cases improving by 2%.

Cons:
- Results on imagnet are not particularly good. The improvement is not significant.
- Why positive-definite precondition matrix rather than positive-semi-definite?
- Section 5: why is a degenerate precondition matrix bad? Fisher and Hessian for deep networks can be highly ill-conditioned.
- Theo 1 seems to have errors. The term M_t in the update rule should show up in the bound on P as an exponential term in the first upper bound.
- Figure 2: On mnist after 20 epochs the model has not reached 1% test error. Not clear if we can make any conclusions from this figure.

After rebuttal:
I keep my rating as weak reject. I reiterate that results look promising. However, the quality and accuracy of the writing are not acceptable for a paper on optimization. In my original review I only named a few problematic statements. I have to clarify that I do not think fixing only those few is enough.

I am also not convinced about the proof of Theorem 1. Basically, section 6 looks very much like section 5 from Baydin et al. 2018. Even the wording is mostly the same. Theorem 5.1 in Baydin et al. 2018 is based on their update rule in Eq 6 in the form of alpha_t = alpha_{t-1} - beta nabla^T nabla, where alpha does not appear in the second term. However, in this paper, the update rule on line 7 in Algorithm 1 is M_t = M_{t-1} + rho * eps *(.) M_{t-1}, where M_t appears in the second term. Hence, the first bound in Theorem 1 in this paper cannot simply be the same as in Baydin et al. 2018.

**Experience Assessment:**

I have read many papers in this area.

**Review Assessment: Checking Correctness Of Derivations And Theory:**

I assessed the sensibility of the derivations and theory.

**Review Assessment: Checking Correctness Of Experiments:**

I assessed the sensibility of the experiments.

**Review Assessment: Thoroughness In Paper Reading:**

I read the paper at least twice and used my best judgement in assessing the paper.

---

> ### Author Response · Authors · 2019-11-08
> **Response to Reviewer 3**
>
> Thank you very much for your constructive comments, as well as noting the strength of the results on CIFAR-10 and robustness to hyperparameter selection! We believe we can address your concerns:
>
> 1. “Results on imagnet are not particularly good. The improvement is not significant.”
>       - We acknowledge that the final difference in test performance is not large (though, as you note, FOP does produce an improvement). However, with any optimizer, accelerated convergence is another axis of potential improvement, and FOP improves upon previous methods (Figure 3) in this regard. We note Zhang et al. (2019) [1] as an example of another optimizer for deep networks whose improvement in generalization is modest in some cases but which reliably accelerates convergence. Moreover, to our knowledge, this is the first application of any hypergradient method to deep DNNs on IMageNet, and we believe that showing such methods can be computationally tractable (and even fairly cheap) at ImageNet scale is an important result in its own right.
>
> 2. “Why positive-definite precondition matrix rather than positive-semi-definite?”
>       - Good question! We should have been more clear. We do not constrain the preconditioner to be positive definite, rather than positive semi-definite. In practice, however, the eigenvalues may be small in magnitude, but are never exactly zero, so the preconditioner is always technically positive definite. Also, a positive definite matrix is generally preferable, as that way no information in the gradient is lost via preconditioning.
>
> 3. “Section 5: why is a degenerate precondition matrix bad? Fisher and Hessian for deep networks can be highly ill-conditioned.”
>       - Apologies for the confusion! We don’t say that we think a degenerate preconditioner is bad, we simply note that it is interesting that the eigenvalues are low enough to render the matrix effectively non-invertible given that  invertibility of the Hessian/Fisher matrices is required for quasi-Newton/natural gradient methods. We believe we have clarified this point.
>
> 4. “Theo 1 seems to have errors. The term M_t in the update rule should show up in the bound on P as an exponential term in the first upper bound.”
>       - Again, apologies for the confusion! Here, we are considering the preconditioning matrix P = MM^T as a whole, not the factorization into M, so M does appear in the bound on the norm of P implicitly. Can you clarify why you believe the first bound should be exponential?
>
> 5. “Figure 2: On mnist after 20 epochs the model has not reached 1% test error. Not clear if we can make any conclusions from this figure.”
>       - Our goal with this figure was to demonstrate the relative performance of FOP for differently-ranked preconditioners in a simple fully-connected network. We do not expect such a fully-connected network to quickly reach 1% test error. While more powerful networks can reach 1% very quickly, doing so even with suboptimal optimizers.
>
> Thank you once again for your helpful comments, and let us know if we can do anything else to further address your concerns!
>
> [1] https://arxiv.org/abs/1907.08610

---

### Official Review · AnonReviewer4 · 2019-11-01
**Official Blind Review #4**

**Rating:** 3

**Review:**

This paper presents a first-order preconditioning (FOP) method to generalize previous work on hypergradient descent to learn a preconditioning matrix that only makes use of first-order information.

Pros:
This paper extends the idea of hypergradient descent in [Almeida et al., 1998; Maclaurin et al., 2015; Baydin et al., 2017] with a preconditioning method. A low-rank FOP is further proposed to lighten the computation burden for the preconditioning matrix.

Cons:
1-	The novelty and contribution is not clear.
2-	The ideas of approximating the preconditioning matrix or factorized approximate inverse have been well studied in the literature, which are not sufficiently cited in the paper, such as Adagrad (Duchi et al. 2011), review in Bottou et al. 2016, etc.
3-	Derivation of Eq.(4) seems to be missing.
4-	Typo errors such as “is can” in page 5.
5-	A mistaken derivation in A.1 Eq.(20). “k” should be “k+1”.

Therefore, I tend to give this paper a Weak Reject score.

**Experience Assessment:**

I have read many papers in this area.

**Review Assessment: Checking Correctness Of Derivations And Theory:**

I assessed the sensibility of the derivations and theory.

**Review Assessment: Checking Correctness Of Experiments:**

I assessed the sensibility of the experiments.

**Review Assessment: Thoroughness In Paper Reading:**

I read the paper thoroughly.

---

> ### Author Response · Authors · 2019-11-08
> **Response to Reviewer 4**
>
> Thank you for your constructive comments! We believe we can address the concerns you expressed. In order:
>
> 1. “The novelty and contribution is not clear.”
>           - We apologize for not more carefully highlighting the differences between FOP and previous algorithms in our initial submission, and believe we have done so in the revised draft:
>           - Previous hypergradient methods which learn a preconditioner (Almeida et al. 1998) restricted it to be diagonal only, which amounts to what is effectively a per-parameter learning rate. FOP learns a matrix with off-diagonal values as well, inducing both scaling and rotation on the gradient.
>           - No previous method has used hypergradients to learn a preconditioning matrix in deep neural networks.
>           - Learning a spatial-only preconditioner for CNNs is, to the best of our knowledge, also novel.
>           - Previous preconditioning methods for deep networks try to approximate specific preconditioners--either the inverse Hessian or inverse Fisher matrix. FOP isn’t learning an approximation to a specific matrix, it simply uses the hypergradient of the loss to directly learn an arbitrary linear transformation of the gradient to accelerate training.
>           - FOP is much more efficient than previous preconditioning methods in deep networks--we believe this is not only such method to be successfully applied to large models trained on ImageNet, but also the first hypergradient method to do so as well.
>           - Our results indicate that the contribution of these changes is an efficient, scalable preconditioner for deep networks that accelerates training and often produces higher generalization performance (Figure 3) in addition to improving the robustness of existing optimizers to hyperparameter selection (Figure 4).
>
> 2. “The ideas of approximating the preconditioning matrix or factorized approximate inverse have been well studied in the literature, which are not sufficiently cited in the paper, such as Adagrad (Duchi et al. 2011), review in Bottou et al. 2016, etc.”
>            - We appreciate these additional citations and have added them to our related work section! However, we also would like once again to stress that FOP isn’t trying to ‘approximate’ anything--this is one of the method’s main sources of novelty. Rather, it is learning an arbitrary transformation online directly from the loss. In particular, we place no constraint enforcing invertibility of the learned preconditioner (see Sections 2.1 and Section 5), which further lightens the computational load relative to other methods.
>
> 3. “Derivation of Eq.(4) seems to be missing.”
>            - Eq. (4) follows directly via the application of the chain rule (Eq. (3)) to Eq. (2), however, we see that this could be confusing, and will include the derivation in the appendix!
>
> 4. “Typo errors such as “is can” in page 5.”
>            - Thank you for pointing this out! We will fix the typos.
>
> 5. “A mistaken derivation in A.1 Eq.(20). “k” should be “k+1”.”
>           - Thank you for catching this as well! We will change it.
>
> Thank you very much again for your comments, many of which have been helpful for improving the paper. We've uploaded a new draft and hope you may have a chance to evaluate it. If you have any follow-up thoughts or questions, we'll also be happy to reply again.

---

### Author Response · Authors · 2019-11-08
**General Response**

We would like to thank all reviewers for your insightful comments and recommendations! We recognize that reviewing is time-consuming work, and we are deeply appreciative. Below, we’ve written responses to each reviewer individually. We have now uploaded a revised version, including:
- a new problem domain: results on reinforcement learning tasks, in which FOP performs even more strongly relative to standard methods than on visual classification (section 4.4 in the text)
- expanded emphasis on novelty and contributions
- additional citations for adaptive optimization methods
- greater clarification of experimental procedures, notation, and theorems
- derivation of the update rule for M
- fixed typos

We hope these changes, in conjunction with our comments below, help address your concerns. Thank you once again for your time!

---

### Decision · Program_Chairs · 2019-12-19

**Decision:**

Reject

**Comment:**

This paper has been assessed by three reviewers who scored it as 3/3/3, and they did not increase their scores after the rebuttal. The main criticism lies in novelty of the paper, lack of justification for MM^T formulation, speed compared to gradient descent (i.e. theoretical analysis plus timing). Other concerns point to overlaps with Baydin et al. 2015 and the question about the validity of Theorem 1. On balance, this paper requires further work and it cannot be accepted to ICLR2020.